# The Associations between 24-Hour Movement Behaviours and Quality of Life in Preschoolers: A Compositional Analysis of Cross-Sectional Data from 2018–2021

**DOI:** 10.3390/ijerph192214969

**Published:** 2022-11-14

**Authors:** Meiyuan Chen, Terence Chua, Zhi Shen, Lee Yong Tay, Xiaozan Wang, Michael Chia

**Affiliations:** 1Physical Education & Sports Science Academic Group, National Institute of Education, Nanyang Technological University, Singapore 637616, Singapore; 2College of Physical Education & Health, East China Normal University, Shanghai 200241, China; 3Department of Astronomy, Shanghai Jiao Tong University, Shanghai 200240, China; 4Centre for Research in Pedagogy and Practice, National Institute of Education, Nanyang Technological University, Singapore 637616, Singapore

**Keywords:** quality of life, physical activity, sleep, sedentary behaviour, preschoolers

## Abstract

Studies show that the quality of life (QoL) of preschoolers is closely related to physical activity (PA), sedentary behaviour (SB), and sleep (SL). Yet many researchers looked at these 24-h movement behaviours as behaviours that are independent of one another when examining the association of QoL with these behaviours. The main purpose of the present study was to describe the temporal trends in PA, SB, and SL in preschoolers and the concomitant association with QoL of children. Annual cross-sectional data on QoL and 24-h movement behaviours of 8045 Singaporean preschoolers were collected from 2018 to 2021. Compositional analysis, linear regression, and isotemporal replacement approaches were used to examine changes in PA, SB, and SL from 2018–2021 and how these changes were associated with QoL. Temporal trends in movement behaviours showed that PA and SL decreased after 2020. During 2018–2021, the association of PA and QoL in preschoolers was characterised by a ‘U’ curve (β_PA-2018_ = 3.06, *p* < 0.001; β_PA-2019_ = 1.43, *p* < 0.05; β_PA-2020_ = −0.43, *p* > 0.05; β_PA-2021_ = 2.82, *p* < 0.001), while SL and QoL were characterised by an inverted ‘U’ curve (β_SL-2018_ = −2.39, *p* < 0.001; β_SL-2019_ = −0.27, *p* > 0.05; β_SL-2020_ = 2.00, *p* < 0.01; β_SL-2021_ = −0.21, *p* > 0.05). SB was significantly and negatively associated with QoL after 2020, with 2020 identified as the inflection point for the change in SB (β_SB-2018_ = 0.67, *p* > 0.05; β_SB-2019_ = −1.16, *p* > 0.05; β_SB-2020_ = −1.56, *p* < 0.01; β_SB-2021_ = −2.61, *p* < 0.01). Using a time re-allocation technique to treat the 2021 data, reallocating time from SB to PA or to SL predicted improvements in QoL of preschoolers (P_all_ < 0.05). The study provided useful information on the temporal trends in PA, SB, SL, and QoL of preschoolers over four years. Additionally, these data provided insights into how changes in QoL are predicted by changes in duration in the 24-h movement behaviours.

## 1. Introduction

Quality of life (QoL) is an important concept and attribute in the field of medicine and public health [1], which is referred to as health-related QoL when applied in the fields of health and disease [2], and is considered a valid health indicator. The Centers for Disease Control and Prevention defines health-related QoL as perceived physical and mental health, including physical and mental health-related functions, of an individual or group over time [3], and this enables the assessment of perceptions of individual function and psychological and social well-being [4]. Health-related QoL refers to how well a person functions in their life. This includes the physical aspect of functioning, such as whether an individual can carry out a range of activities of daily living without much difficulty, and the social aspect of their functioning (for example, the extent to which he or she is able to interact with family and friends or keep up with their peers in school ((social and school functioning)). Health-related QoL also includes the well-being of a person such as the extent to which he or she expresses and manages his or her emotions appropriately (emotional functioning) [5]. Monitoring and tracking of health-related QoL facilitate the identification of people with poor health and allow for interventions to improve health. Additionally, the exploration of influencing health factors provides a basis for improving QoL.

Health-related QoL is also central to paediatric-related practice [6], and various types of child QoL are a hot topic of research in the field of paediatrics [1,2,4]. For instance, a series of studies show robust relationships between the QoL of preschoolers and their physical activity (PA), sleep (SL), and sedentary behaviours (SB). The World Health Organization (WHO) Guidelines on PA, SB, and SL for children under 5 years of age suggest that improving PA, SL, and reducing SB in children will contribute to improving children’s physical health, mental health, and well-being [7,8]. SB refers to any waking behaviour characterized by an energy expenditure ≤ 1.5 metabolic equivalents (METs), while in a sitting, reclining or lying posture SB can be in forms of non-screen-based and screen-based (e.g., using a smartphone/tablet) [9]. The Commission on Ending Childhood Obesity has also emphasized the importance of the interaction between PA, SB, and adequate SL for physical and mental health and well-being [10].

Previous studies reveal that increasing PA improves physical functioning, psycho-emotional functioning, and overall QoL at all ages [11]. PA is significantly and positively associated with children’s QoL [12], and more PA in children is associated with better levels of QoL. There is also a significant dose–effect relationship between children’s SB and health, such as lowering sedentary time leads to reductions in body mass index (BMI) over a period of time [13,14], and reducing SB is beneficial for improving children’s psychosocial health [12]. Some scholars examined the relationship between PA and SB with QoL and concluded that increasing PA and reducing SB can effectively improve QoL [15]. For SL, the American Academy of Sleep Medicine advocates that adequate SL is effective in promoting QoL, physical health, and mental health in children [16]. As sleep insufficiency is quite common in school-going aged children, getting more SL within the age-specific recommended range of sleep hours is associated with higher QoL [12]. Indeed, behavioural problems attributed to deficient sleep is negatively associated with QoL [17].

Current international policy and research trends emphasize the important influence of PA, SB, and SL on QoL. It is noteworthy that PA, SL, and SB make up a 24-h day of interdependent movement behaviours, and an increase in the time spent on one behaviour could necessarily lead to decreases in other behaviours [18]. Even though some scholars have examined SL and PA in their research, or linked SB to PA [12,15,19], as with other studies, the accomplished research is essentially a single analysis of a routine behaviour that ignores or disregards the interactions between 24-h behaviours that could limit the health benefits of behavioural optimization [20]. Recently, compositional data analysis (CoDA) has become popular, as the technique takes into account all components of a whole simultaneously by a set of logs, while being able to circumvent some common problems in traditional statistics (e.g., covariance) and also investigates the association between different temporal components and health outcomes through modeling [18,21,22]. A few studies have examined the relationships between 24-h behaviours and health outcomes using CoDA [23,24,25,26,27,28,29]. Even then, apparently all research studies based on CoDA are single-year cross-sectional studies that focused on children and adults, with relatively fewer focusing on preschoolers.

Therefore, the present study has three aims. They are:(1)To investigate the changes in 24-h behaviours (PA, SB, and SL) of preschoolers from 2018–2021 using CoDA;(2)To examine the trends of association between 24-h behaviours and QoL of preschoolers from 2018–2021 based on linear regression models;(3)To predict changes in QoL by re-allocating time between PA, SB, and SL behaviours using an isotemporal substitution analysis model with PA, SB, and SL behaviours from data collected in 2021.

In the accomplishment of these objectives, the use of CoDA over a period of four years offers an alternative to the traditional research approach of examining the relationship between 24-h behaviour and quality of life, thereby offering fresh insights to the relationships between preschoolers’ PA, SB, SL, and QoL. Additionally, the isotemporal substitution model provides useful information about the quantum of benefit for substituting one behaviour for another.

## 2. Materials and Methods

### 2.1. Design and Participants

The study is based on the International iPreschooler Surveillance Study Among Asians and otheRs (IISSAAR), a project that investigates preschoolers’ digital and non-digital behaviours outside of school, including digital media use, non-digital media SB, SL, PA, and QoL. Repeated cross-sectional parent-reported survey responses were collected from 2018 to 2021. Inclusion criteria were parents with preschoolers aged 2 to 5 years old enrolled in one of the anchor operator preschools or kindergartens in Singapore. These preschool operators operated under a private–public enterprise arrangement called the Anchor Operator Scheme [30]. The private–public arrangement provides good quality early childhood education to the masses and especially to lower income and disadvantaged households with financial subsidies. A research invitation letter inviting parents to participate in the study was sent out by the preschool operators. Parents provided informed consent to their voluntary participation and were assured of anonymity of their survey responses.

Cross-sectional parent-reported survey responses were collected from 2018 to 2021, and the periods of data collection in each year are as follows: For year 2018, data collection were between the periods of March and August. For year 2019, data collection commenced in March and ended in September. For year 2020, data collection was deferred to August due to the emergence of COVID-19 at the beginning of the year. Completion of data collection was in November that year. For year 2021, data collection commenced in April and concluded in July.

Eligible parents who gave their informed consent were directed to an online questionnaire package that comprised the Surveillance of digital Media hAbits in earLy chiLdhood Questionnaire (SMALLQ^®^) and Pediatric Quality of Life Inventory (PedsQL™). A total of 9069 complete responses were collected, of which 8045 (88.7%) were retained after further data cleaning. The data sorting process is detailed in Figure 1 of Appendix A. Ethics considerations of the research was approved by the ethics committee of the Nanyang Technological University (IRB 2017-09-036 and IRB 2019-02-036).

### 2.2. Measurements

#### 2.2.1. Assessment of the PA, SL and SB

Researchers in studies that employ compositional data analysis (CoDA-related) commonly use accelerometers for the data collection on PA and SB, but this approach and concept are equally applicable to subjective reporting of time-use [19,28]. The SMALLQ^®^ was used in the present study to garner temporal data on PA, SL, and SB in children aged 2–5 years. The development, validity, and reliability of SMALLQ^®^ are elaborated.

##### Surveillance of Digital Media in Early Childhood Questionnaire (SMALLQ^®^)

The SMALLQ^®^ is not a psychometric assessment tool. Rather, it was developed to solicit, in particular, information from adult parents about the digital (screen time, type, and purpose) and non-digital media (indoor and outdoor play, sleep duration, and quality) habits of their preschool-aged children on a weekday and on a weekend. Parents were also queried about their attitudes and concerns towards child digital media use and their awareness and practice of child media use guidelines. SMALLQ^®^ is made up of three segments: (I) Digital media use by parent and child, (II) Child’s non-digital behaviours, and (III) Child and parent information [28]. A copy of the questionnaire package is appended as Appendix A.

##### Development and Front-Loaded Validity of the SMALLQ^®^

Multiple stakeholders were involved in the development of the SMALLQ^®^—these were three experts in early childhood, an expert in construction of online questionnaires, three preschool teachers, and five parents with children aged below 6 years old, in accordance with the guidelines promoted by Boynton and Greenhalgh [31]. The development of SMALLQ^®^ was guided by the seven-step AMEE framework—Guide No. 87 [32]. The seven-step framework has been used to develop high-quality surveys that are suitable for research and programme evaluation. Questions on digital media use in SMALLQ^®^ were framed to solicit information particularly about the content (e.g., is the screen media for education or for entertainment; is the screen media age-appropriate), context (e.g., whether parent is present with child when screen media is in use; how is screen media delivered: computer, smart phone, game console), and dose (e.g., how long is screen media used by parent and by the child) of digital media use based on the conceptual understanding of screen media parenting proposed by O’Connor and his colleagues [33]. Five parents with children aged below 6 years old who were not involved in the research provided input on all items in the draft questionnaire and the cognitive load of the questionnaire. Based upon feedback, some questionnaire items were re-worded or re-ordered, thereby front-loading the validity of the SMALLQ^®^.

##### Established Face and Content Validity of the SMALLQ^®^

The questionnaire items in SMALLQ^®^ were then face- and content-validated by four independent experts based upon the criteria of representativeness, clarity, and relevance [32]. Thereafter, the questionnaire was hosted on an online survey platform called Qualtrics^®^XM and was pre-tested on 137 parents of preschool children. None of the parents reported any difficulty in answering or responding to the questionnaire items. The development and validity of SMALLQ^®^ was described previously by the authors [28].

##### Internal Consistency (Reliability) of the SMALLQ^®^

The Cronbach values for PA over the four years were as follows: 0.808 for 2018, 0.814 for 2019, 0.776 for 2020, and 0.801 for 2021. The Cronbach values for SB over the four years were as follows: 0.889 for 2018, 0.863 for 2019, 0.846 for 2020, and 0.766 for 2021. The Cronbach values for SL over the four years were as follows: 0.733 for 2018, 0.717 for 2019, 0.642 for 2020, and 0.576 for 2021. A Cronbach’s alpha of 0.70 is commonly cited as an acceptable threshold [34]. Therefore, there was modest acceptable internal consistency for PA and SB across the four years. For SL, internal consistency was modest in 2018 and 2019 and fairly modest for the last two years.

##### Classification of Activities into PA, SB, and SL

Parents were asked in the questionnaire to recall the durations of different activities of their children. An example of a recall question on time spent on physical activity is: ‘In the last 7 days, how much time did your child spend (in a day) on outdoor physical play (e.g., playing’ hide-and-seek’ or ‘catching’ in a playground)?’ An example of a recall question on time spent on SB is: ‘In the last 7 days, how much time did your child spend (in 1 day) using digital media for education/learning (e.g., reading digital e-book, learning math via educational apps).’ An example of a recall question on sleep is: ‘In the last 7 days, what was the average duration of your child’s night-time sleep?’.

The first stage of data curation was categorizing the durations reported on different activities that were solicited from SMALLQ^®^ into PA, SB, and SL. Helping with household chores, indoor play, and outdoor play were grouped as PA. On-screen activities (e.g., learning educational content on screen media devices and watching television) and non-digital media sedentary activities (e.g., doing craftworks, drawing, and reading printed books) were classified as SB. Naps and night-time sleep were grouped as SL (as detailed in Appendix A). Thereafter, durations of activities that fell under the same corresponding categories were added up to derive PA, SB, and SL durations.

Following that, we determined the range of SL durations for different age groups based on the WHO Guidelines on PA, SB, and SL for children under 5 years of age [8] and the American Academy of Sleep Medicine [35]. Simultaneously, SL was used as an anchor variable to determine the time categories of PA and SB for data cleaning, where the total time of the three behaviours (PA, SB, and SL) add up to 24 h. The data filtering criteria and procedures are elaborated in Appendix A.

#### 2.2.2. Assessment of the Quality of Life

The Pediatric Quality of Life Inventory (PedsQL™) is a widely reported health-related quality of life (HRQoL) instrument, and the acute 7-day recall parent-reported version was used [36]. The parent-reported questionnaire has 21 items for parents with children aged 2–4 years and 23 items for parents with children aged 5–6 years and includes general health, physical health, and psychosocial health, of which physical health includes physical functions (8 items), and psychosocial health includes emotional functions (5 items), social functions (5 items), and school functioning (3 or 5 items). Parents were asked to rate how much of a difficulty each problem was for their children related to physical functioning, emotional functioning, social functioning, and school functioning. Examples for physical functioning include child walking more than a block, running, lifting something heavy, or having a low energy level. Examples for emotional functioning are feeling afraid or feeling angry. Examples for social functioning are getting along with other children or keeping up when playing with other children. Examples for school functioning are paying attention in class or missing school because of illness. The answers to each entry are divided into five levels from 0–4 (with 0 being ‘never’ to 4 being ‘almost always’), with the corresponding scores to be converted into 100, 75, 50, 25, and 0. Each dimension’s score is the sum of the scores of the subordinate items divided by the number of items completed, and the total score is the combination of the scores of each item divided by the number of entries received for the whole scale.

In addition, the QoL questionnaire was divided into several versions according to age groups, such as 2–4 and 5–7 years. Because of the age of preschoolers, the parent proxy report version was chosen for the survey. At the same time, the question “which age group does your child belong to” was used to triage parents, and the corresponding age group questions were presented to ensure the validity of the data entered. The scale is currently used in several countries around the world to monitor children’s health, with Cronbach alpha values for the full 23-item scale approaching 0.90 for self- and proxy-report [37]. Specifically for the present study, the PedsQL composite scores (physical health score, psychosocial health score, and total health score) yielded very good reliability with Cronbach alpha values of 0.92, 0.91, 0.91, and 0.89 for 2018, 2019, 2020, and 2021, respectively.

#### 2.2.3. Treatment of Data

The data cleaning process is outlined in Appendix A. The number of null values for QoL, age, and gender were less than 5% of the total data. Null values for QoL and age were replaced by the mean values of the corresponding age groups, and the null values for gender were replaced by mode for gender [38]. Following data cleaning, the final dataset across four years used for statistical analysis numbered 8045.

#### 2.2.4. Covariates

Both gender (male or female) and age of child were set as covariates for controlling its potential influence in the study analysis.

### 2.3. Statistical Analysis

#### 2.3.1. Application of Compositional Data Analysis (CoDA)

Compositional data analysis (CoDA) is a well-established statistical method that is widely used in research where multiple “components” (variables) are involved [39,40]. The method considers that all “components” of a whole have mutuality where the scale of the component data are constant, i.e., the relative ratio between components is constant, and they are naturally subject to a unit-sum constraint [18,39]. For instance, if the D-dimensional vector X = [x1, x2, …, xD], then its components satisfy the overall constraint [16]. In epidemiology, the CoDA application is then the logarithm of ratios of the time-use behaviours [21]. This is an important part of analysing interdependencies between components or correlations with other outcome indicators. The transformation of the log-ratios of different components is an important process. Moreover, the isometric log-ratio (ilr) is the recommended data transformation method that allows for discrete trend analysis of data based on CoDA and also the construction of linear regression models and isotemporal substitution models [18,21].

In recent years, compositional analysis has been used to treat data emanating from several countries, such as time spent on various types of behaviours per day and their impact on health through relative changes in the time spent in these different behaviours [21]. Using CoDA to examine lifestyle habits or behaviours provides a more comprehensive and integrated approach to exploring the impact of 24-h behaviours on health [23]. Hence, CoDA was applied for data analysis in the present study. The specific analysis principles are detailed in Appendix A.

#### 2.3.2. Analysis Process

Besides conducting a demographic analysis of PA, SB, and SL of preschoolers in 2018–2021 using ANOVA based on age, gender, and ethnicity, descriptive statistics (such as compositional mean and variation matrix) based on the features of compositional analysis were also generated [21,22,41]. After standardizing and normalizing PA, SB, and SL, the central values of PA, SL, and SB were calculated separately for preschoolers from 2018 to 2021, and the significance of differences was calculated using ANOVA. Further ternary plots with PA, SB, and SL were generated to compare the trends of changes in preschoolers’ 24-h behaviours over the four years. The variation matrix was also calculated by combining the variances of the logarithms of all pair-wise ratios between parts of PA, SB, and SL behaviours to assess the dispersion and structure of the relative dispersions. The closer their proximity is to the value 0, the higher the interdependence is between the two behaviours. At the same time, uniform quantiles were used to combine the QoL scores for classification, and compositional geometric mean bar plots of 24-h behaviours for different levels of QoL in each year were produced, to compare the changes in PA, SB, and SL of preschoolers at different levels of QoL that corresponded to the QoL for each year.

During the application of compositional analysis, the temporal component for each behaviour (PA, SB, or SL in each case) was used as the independent variable and QoL as the dependent variable. Subsequently, two models were generated by using an isometric log-ratio (ilr) data transformation (Appendix A): (1) We developed a linear regression model for 2018–2021 and adjusted the model for gender and age variables. We then created predicted-change graphs to compare the trend of these associations. Linear regression assumptions were checked for linearity and normality of distributions. (2) Based on the adjusted linear regression model, isotemporal substitution models were further constructed employing data from 2021, with an hour of behavioural times (PA, SB, and SL) reallocated in 15-min increments. For each pair-wise combination of time reallocation, the difference in predictions was calculated, and the 95% confidence intervals with significance were calculated. This was calculated to investigate the dose–effect between a 24-h behavioural change and QoL in preschoolers.

In the process of data analysis, the demographic analysis was completed through SPSS Version 26.0 (IBM, New York, NY, USA), and all other data associated with the compositional analysis were completed through the R program Version 4.2 () (R Core Team, https://www.r-project.org/, accessed on 1 October 2022) [42]. For all statistical analysis, the level of significance was accepted at *p* < 0.05.

## 3. Results

### 3.1. Descriptive Statistics

The descriptive characteristics of the cross-sectional study sample from 2018–2021, and the difference of the characteristics across 4 years are presented in Table 1. A total of 8045 parents of preschoolers participated in the study (N_2018_ = 2677, N_2019_ = 1961, N_2020_ = 2403, N_2021_ = 1004). The number of boys and girls among preschoolers in each year of study was relatively even at around a 50% split for each gender; for the covariates, it showed significant differences in mean age (P_age_ < 0.001) across 4 years, but not for gender (P_gender_ > 0.05). For 24-h behaviours (PA, SB, and SL), there were significant differences in SL and SB (P_SL_ < 0.001, P_SL_ < 0.001) across 2018 to 2021, but not for PA (P_PA_ > 0.05) in the 4-year data sample. The relative ethnic distribution of parent respondents in the present study is somewhat similar to the ethnic make-up of the population in Singapore (where Chinese is the majority race, followed by Malay, Indian, and Eurasian). Across 2018 to 2021, 65.9% of parent respondents were Chinese, 16.7% were Malay, 11.2% were Indian, 0.3% were Eurasian, and 6.0% indicated others.

To better understand the 24-h behavioural changes in children from 2018–2021, compositional analysis was used to process the data. The CoDA helps us to understand the data more spatially, and the ternary plot represents this characteristic well [43]. Analysis of variance revealed that the time composition for PA, SB, and SL showed significant differences across the four years (F_PA_ = 7.42, *p* < 0.001; F_SL_ = 5.61, *p* < 0.001). The trends in 24-h behaviours for PA, SB, and SL of preschoolers from 2018–2021 are illustrated as ternary plots in Figure 1. Each point, which is the compositional mean, corresponds to the central value of PA, SB, and SL for the corresponding year. As an overview, it seemed that SB gradually increased in 2020 and 2021 compared to 2019 and 2018, and both PA and SL gradually decreased. But in 2021, there was an increase in physical activity among preschoolers.

The variability of PA, SB, and SL data is displayed in Table 2 in the variation matrix, which contains all pair-wise log-ratio variances. In terms of any individual year in 2018–2021 or overall, the smallest pair-wise log ratio variances were for PA and SL (*log*_PA/SL-2018_ = 0.33, log_PA/SL-2019_ = 0.38, log_PA/SL-2020_ = 0.37, log_PA/SL-2021_ = 0.36, log_PA/SL-2018–2021_ = 0.36), the highest was for PA and SB, followed by SL and SB. When the variance of log (PA/SL) is closest to zero in comparison to the other pairs of behaviours, it is indicative that the proportional relationship or interdependence of behaviours is highly present. In the present study, this interdependence is largest between PA and SL and is smallest between PA and SB.

### 3.2. Composition of a 24-h Day in PA, SB, and SL by Different QoL Quantiles

To gain insights into the PA, SB, and SL for preschoolers with different QoL from 2018 to 2021, we grouped QoL into quantiles and calculated the absolute proportions of different time-use in PA, SB, and SL. These are illustrated as standard bar plots in Figure 2. Overall, preschoolers with higher QoL were more physically active (had greater PA) and less sedentary (had lesser SB), regardless of the year. Further analysis revealed that this observation was more pronounced in 2018 and 2019 and fluctuated slightly in 2020 and 2021. Our data also showed that the higher the QoL of children, the longer they slept (greater SL) in 2020, but this observation of SL with QoL was not evident in 2021.

### 3.3. Compositional Linear Regression Models: Associations between 24-h Behaviors and QoL

The compositional linear regression models for the cross-sectional association analyses for the years 2018–2021 are presented in Table 3. After adjusting for age and sex, the results in Table 3 showed that there were significant associations between weekend 24-h behaviours and QoL in preschoolers in 2018, 2019, 2020, and 2021 (P_all years_ < 0.001), respectively, with the explanatory power of 7%, 4%, 3%, and 3%. Specifically, the associations between PA, SB, SL, and QoL varied from year to year. In 2018, preschoolers’ PA time was positively and significantly correlated to QoL, while SL was negatively and significantly correlated to QoL (β_PA_ = 3.06, *p* < 0.001; β_SL_ = −2.39, *p* < 0.001). In 2019, preschoolers’ QoL was positively and significantly correlated to PA time (β_PA_ = 1.43, *p* < 0.05), but not SB or SL. In 2020, preschoolers’ SL time was positively and significantly correlated to QoL, and SB was negatively correlated to QoL (β_SL_ = 2, *p* < 0.01; β_SB_ = −1.56, *p* < 0.01). In 2021, PA of preschoolers showed a positive and significant correlation to QoL again, and SB of preschoolers showed a negative and significant correlation to QoL (β_PA_ = 2.82, *p* < 0.001; β_SB_ = −2.61, *p* < 0.001). It is important to note that the associations and significance of each stated behaviour (PA or SB or SL) correlated to QoL are relative to the other two behaviours.

The changes in estimating beta values for PA, SL, and SB and QoL of each model over the four years are also shown as trend graphs in Figure 3. Year 2020 was the turning point where the association between PA and QoL in preschool children over four years is depicted as a “U-shaped” curve, while the trend of SL in preschool children is depicted as an inverted “U”-shaped curve (Figure 3).

Firstly, the association between PA and QoL for preschoolers showed a decreasing trend from 2018 to 2020, with the weakest association in 2020, and the trend changing from positive to negative, and the trend turning positive again in 2021 (β_PA-2018_ = 3.06, *p* < 0.001; β_PA-2019_ = 1.43, *p* < 0.05; β_PA-2020_ = −0.43, *p* > 0.05; β_PA-2021_ = 2.82, *p* < 0.001). 

Secondly, the association between SL and QoL in preschool children showed an increasing trend from 2018 to 2020, peaking in 2020, with the association turning from negative to positive, and the trend declined again in 2021, with the association turning negative (β_SL-2018_ = −2.39, *p* < 0.001; β_SL-2019_ = −0.27, *p* > 0.05; β_SL-2020_ = 2.00, *p* < 0.01; β_SL-2021_ = −0.21, *p* > 0.05).

Thirdly, the negative association between preschoolers’ SB and the QoL increased consistently and was significantly different between 2020 and 2021 (β_SB-2018_ = 0.67, *p* > 0.05; β_SB-2019_ = −1.16, *p* > 0.05; β_SB-20220_ = −1.56, *p* < 0.01; β_SB-2021_ = −2.61, *p* < 0.01).

### 3.4. Isotemporal Substitution Analysis: Effect of Time Re-Allocation

The data and trends of the isotemporal substitution model in 2021 are presented in Table 4 and Figure 3, respectively. Our results showed that preschoolers’ QoL decreased by 0.2%, 0.3%, 0.6%, 0.8% and 0.3%, 6%, 1.0%, 1.2%, respectively, when reallocating time for PA to SL and SB in increments of 15 min, for up to 60 min. On the other hand, preschoolers’ QoL increased by 0.2%, 0.3%, 0.5%, 0.6% and 0.3%, 0.6%, 0.9%, 1.2%, respectively, when reallocating time for SL and SB to PA. The changes in preschoolers’ QoL resulting from the reciprocal reallocation of behavioural time are not symmetrical or linear. In addition, preschoolers’ QoL decreased significantly by 0.1%, 0.3%, 0.4%, and 0.5% when reallocating 15–60 min of SL to SB, and increased significantly by 0.1%, 0.3%, 0.4%, and 0.6% when SB was reallocated to SL. The changes in QoL were significantly different in all conditions (all *p* < 0.05). From Figure 3, it is observed that in reallocating more time in increments of 15 min, the change in QoL is correspondingly greater.

## 4. Discussion

This study explored the changes in 24-h behaviours of preschoolers and their associations and trends with QoL in 2018, 2019, 2020, and 2021 by using CoDA. To the authors’ knowledge, this is apparently the first study to apply compositional analysis to the combined behaviours of PA, SB, and SL as factors affecting the QoL in preschoolers and plausibly the first study to examine changes in the relationships between 24-h behaviours and QoL in preschoolers using cross-sectional data over four consecutive years. Several seminal findings in the present study are worthy of discussion

### 4.1. Trends in 24-h Movement Behaviors of Preschoolers over 2018–2021

Our study showed that PA and SL of preschoolers were more closely interdependent on each other than SB. After 2020, SB in preschoolers increased significantly and SL and PA decreased significantly, but in 2021, PA increased, and SB decreased in preschoolers. In the context of Singapore, this was most likely due to the COVID-19 epidemic. This explanation is affirmed by research that shows that preschoolers’ health behaviours changed as a result of epidemic protection policies following the COVID-19 outbreak in 2020, as well as a significant decrease in PA and a significant increase in SB in the United States, Poland, and Canada [45,46,47,48]. School and office closure policies were implemented in Singapore in 2020 for a specific period called a circuit breaker (CB). The CB was a partial national lockdown that was implemented between 4 April and 2 May 2020, to contain the spread of the COVID-19 virus (MOH, 2020). For instance, during the CB, a study on Singaporean children aged 3–16 years showed that children and adolescents slept longer in 2020 during the CB [49], and this is contrasted with the present result where preschoolers slept less in 2020. A major reason for this observed difference was that the cited study involved older children and adolescents, where the adolescents reportedly slept 0.7 h more during the CB when schools were closed, and both adults and young people stayed home [49]. In 2021, as Singapore sought a path to coexist with COVID-19, the policy was relaxed in a calibrated manner after June (MOH, 2021), and public places opened one after another, and school and extracurricular activities were resumed (MOE, 2021). Overall, this may have led to a relative increase in PA and a relative decrease in SB in children. The present data for years 2020 and 2021 were collected outside of the CB period in Singapore, where movement curtailments would have been less stringent.

### 4.2. PA, SB and SL in Relation to QoL and Impact of the COVID-19 Epidemic

Our study showed that the more physically active preschoolers were, the shorter the duration was of SB, and the higher the QoL was of preschoolers. For instance, the PA of preschoolers was significantly and positively associated with QoL in all years except 2020. At the same time, the SL of preschoolers was significantly and negatively associated with QoL in 2018, but significantly and positively associated in 2020, while SB of preschoolers was significantly and negatively associated with QoL in 2020 and 2021. Several studies have shown that increased PA has a positive effect on children’s motor development, executive function, and healthy growth [50,51,52], and increased PA is ‘good medicine’ for QoL improvement, whether for a normal child, an obese child, or a child with other diseases [53,54,55,56]. The results of our study affirmed these ‘wisdoms’ among preschoolers. However, for the year 2020, PA in preschoolers did not present a significant association with QoL, while SL and SB in preschoolers presented a significant association with QoL. These findings could plausibly be explained by the period before (2018 and 2019) and the period during (2020 and 2021) the COVID-19 epidemic. Many studies on the COVID-19 epidemic showed that children’s behavioural patterns changed due to home isolation or participation in online courses, such as a decrease in outdoor exercise and an increase in screen use behaviour, and these behavioural alterations seriously affected children’s health behaviours and QoL [57,58]. There is an inverse relationship between SB and health benefits [15], and an increase in SB will increase health risks [13]. In the present study, the SB of preschoolers after 2020 was significantly increased, plausibly because of the COVID-19 epidemic, so the negative and significant associations between SB and QoL were presented in 2020 and 2021.

Our study also showed that preschoolers may first need to ensure adequate SL as opposed to PA to ensure healthy physical and mental development during the period of the COVID-19 epidemic in 2020. The importance of adequate SL in early childhood cannot be disputed. Early childhood is a critical period for the development of self-regulatory abilities (e.g., attention regulation, emotion regulation, etc.) [59,60], increased SL can effectively improve brain connectivity [61], and adequate SL plays a crucial role in the development of physical health and social-emotional functioning of young children [1,2,3,4]; children with insufficient SL will have a poorer QoL [16,17]. Therefore, preschoolers needed more sleep for self-regulation to improve their quality of life in 2020 when the health and hygiene environment was significantly altered because of the COVID-19 pandemic. However, it should be noted that there is also a ‘U-shaped’ relationship between SL and QoL, and excessive SL has adverse effects [62,63,64], while the ternary plot (Figure 1) showed that preschoolers slept significantly longer in 2018 than in 2020. This might plausibly explain our finding of a negative correlation between SL and QoL in preschoolers in 2018. However, this is based on the results of compositional analysis, and it is likely that sleep duration is altered by the normalization of 24-h movement behaviours, but further research is needed to confirm this finding.

### 4.3. Effect of Time Re-Allocation in PA, SB and SL on QoL

From a compositional data analysis perspective, even though SL in preschoolers increased in 2021 relative to 2020 in the present study, the increased SL was not significantly associated with QoL, but it is plausible that SL ‘acted as a as a booster’ to maximize the health benefits of PA. As shown in the isotemporal substitution model of 2021 (Figure 4), preschoolers’ QoL significantly decreased when the time spent on SL was reallocated to PA in 15-min increments up to 60 min, but significantly increased when the time spent on SB was replaced by 15, 30, 45, and 60 min of SL. Correspondingly, the QoL of preschoolers also increased significantly when the time spent on SB was substituted by PA in 15-min increments up to 60 min (Table 4).

Findings from the present study therefore reaffirmed that both PA and SL are important indicators of healthy child development [65], and that both SL and PA play important roles in QoL of preschoolers in different environments. Further, under the current environment of the COVID-19 epidemic, parents, teachers, and caregivers of preschoolers should help preschoolers to increase PA, have sufficient SL, and reduce SB to improve QoL.

### 4.4. Strengths and Limitations

The importance of PA, SB, and SL as lifestyle behaviours in young children have come to the fore. The World Health Organization and national bodies have published guidelines for these lifestyle behaviours. Mainly, these guidelines provide daily recommendations for PA and SL for children and emphasize a reduction of SB [8,66,67,68]. In line with these important recommendations, several studies on young children have examined the effects of SL and SB or PA and SB, from a single dimension or combined, on children’s QoL [69], and even when the three behavioural combinations were analysed, only the best and worst combinations of behaviours were compared (e.g., high PA/high SL/low SB versus low PA/low SL/high SB), and no intermediate combinations were compared (e.g., low PA/high SL/low SB) [70]. The present study addressed some of the limitations of previous studies and demonstrated the importance of PA in preschoolers in the current COVID-19 environment and affirmed the positive and supportive role of SL. At the same time, the negative impact of SB is foregrounded and affirmed by others [71]. These data provide a useful reference for future research for the maintenance or improvement of wellbeing of preschoolers. In this study, we presented new insights with the use of CoDA by explaining the integrated relationships between PA, SB, SL, and QoL and temporal changes in these behaviours and their relationship to health.

The present study has several limitations. Foremost, in order to obtain a larger sample size, the data on preschoolers’ PA, SB, and SL and QoL were all parent-reported using questionnaires, where parent responses were subjective and prone to recall bias and social desirability bias. Nonetheless, recall bias was minimized by having parents recall PA, SB, and SL and QoL of their children over a period of 7 days. Social desirability bias was also minimized by assuring parents of their anonymity. Another limitation is that data collected in 2020 was at a different time point of the year compared with data collected in 2018, 2019, and 2021. It is not known if there are temporal variabilities in PA, SB, and SL of preschoolers’ issues that remain unaddressed and await future research. Nonetheless, future research on the 24-h behaviours of PA, SB, and SL should employ a combination of subjective (appropriately developed questionnaires) and objective (such as accelerometers) methods [72,73] for a more complete understanding of 24-h movement behaviours in preschoolers.

### 4.5. Future Directions

As the present study involved cross-sectional data of lifestyle behaviours of preschool children and their QoL over four consecutive years in 2018, 2019, 2020, and 2021, only associations among the variables could be elucidated and cause-and-effect explanations remain elusive. The causal effects of 24-h movement behaviours and preschoolers’ QoL need to be further explored with longitudinal studies. Future studies on different forms of non-screen SB and light physical activity, such as eating during family mealtimes or sitting during transportation, that occur during a 24-h day and not addressed in the present study, are worthy of research attention.

## 5. Conclusions

Overall, 24-h behavioural changes in PA, SB, and SL in preschoolers were significantly associated with their QoL, but the specific association varied depending on the different healthy and hygienic environments across the four years, with the PA and SL of preschoolers decreasing and SB increasing, relatively, after the COVID-19 outbreak in 2020. Between 2018 and 2021, the QoL of preschoolers showed a positive “U-shaped” curve in relation to PA and an inverted “U-shaped” curve in relation to SL, with 2020 as the turning or inflexion point, and SB also showed a significant negative association with QoL after 2020. It appeared that in the present environment (i.e., living in the COVID-19 epidemic), safeguarding PA, increasing SL, and reducing SB in preschoolers is an effective approach to improve the QoL of preschoolers.

## Figures and Tables

**Figure 1 ijerph-19-14969-f001:**
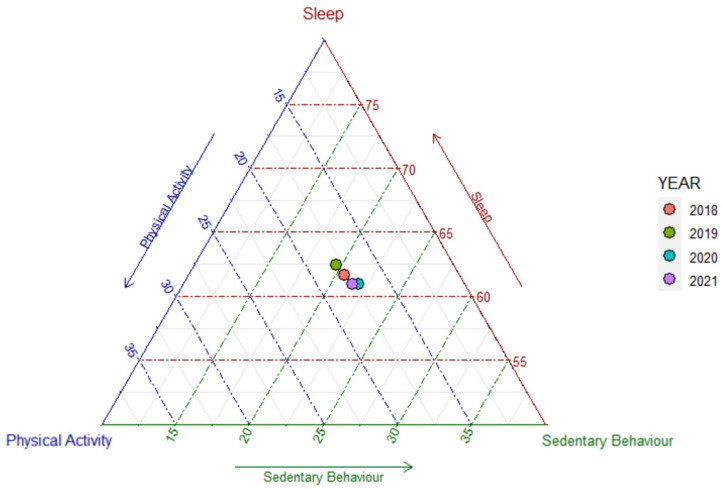
Ternary plot of the sample composition of time spent in PA, SB, and SL from year 2018–2021. Different coloured points represent the different years, the red point represents 2018; the green point represents 2019; the blue point represents 2020; and the purple point represents 2021.

**Figure 2 ijerph-19-14969-f002:**
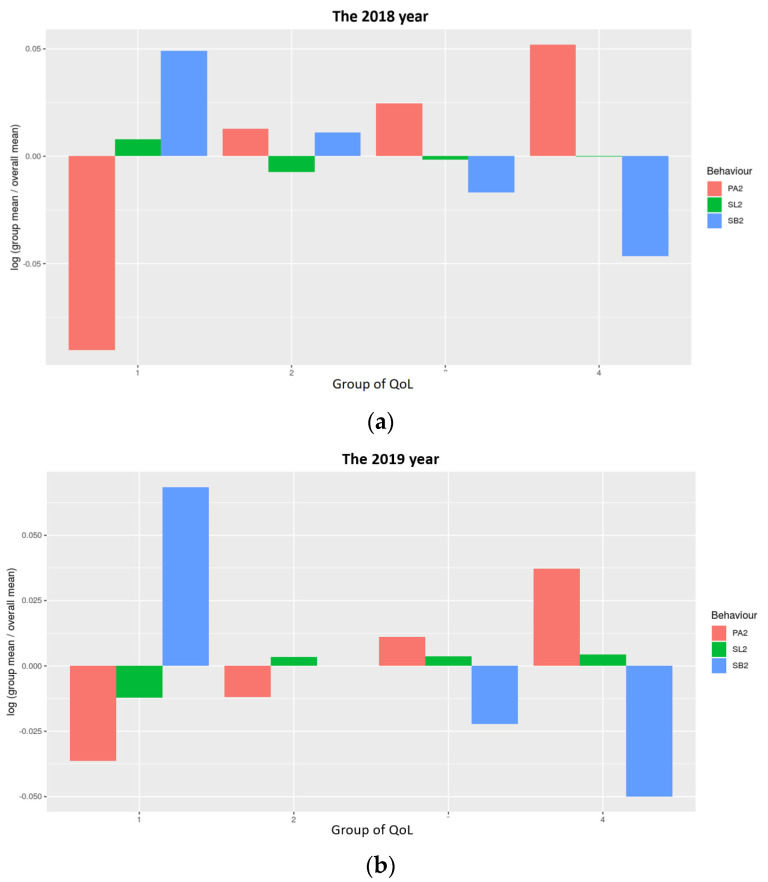
Composition of a 24-h day in PA, SB, and SL grouped by 1–4 Qol quantiles as compositional data analysis of the relative importance of the group mean time spent in PA, SB, and SL with respect to the overall compositional mean time from 2018 to 2021 (**a**–**d**). The *y*-axis displays the log-ratio value of compositional means by group (and overall) while the *x*-axis shows the quality of life (QoL) quantiles.

**Figure 3 ijerph-19-14969-f003:**
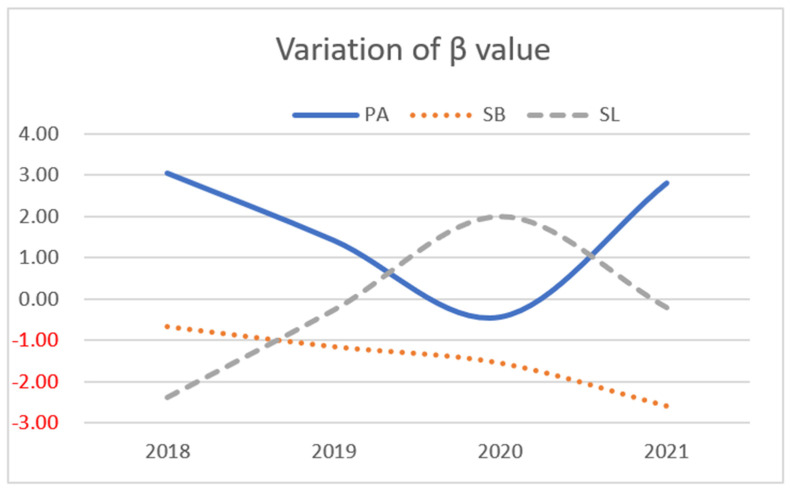
A line graph of the regression coefficients for PA, SB, and SL, respectively, plotted across four years. β values are regression coefficients and were chosen from the adjusted linear regression model, which represents a total effect of the predictor variables [44].The *y*-axis displays the β value, while the *x*-axis shows the different years from 2018–2021.The different lines represent different changes in β for PA, SL, and SB. PA = Physical activity; SL = Sleep; SB = Sedentary behaviour.

**Figure 4 ijerph-19-14969-f004:**
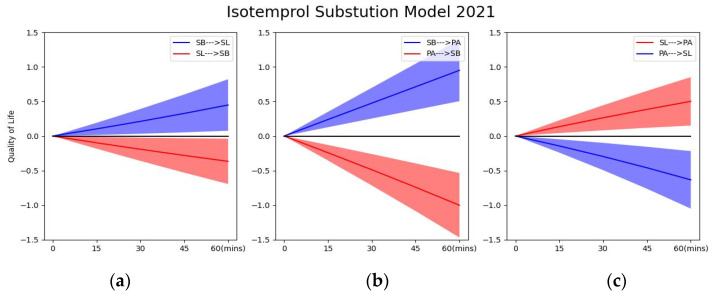
The graphs show the trends in QoL changes after the isotemporal replacement of preschoolers’ 24-h behaviours in 2021. (**a**) presents trends in preschoolers’ QoL after time reallocation for SL and SB; (**b**) presents trends in children’s QoL after time reallocation for PA and SB; (**a**,**c**) presents trends in preschoolers’ QoL after time reallocation for SL and PA; the 95% confidence interval is also presented in the shaded part of the image.

**Table 1 ijerph-19-14969-t001:** Descriptive characteristics of the cross-sectional study sample from 2018–2021.

Variable	Category	Years				*p*
		**2018**	**2019**	**2020**	**2021**	
n		2677	1961	2403	1004	
Age (yr)		3.77 (1.25)	3.79 (1.27)	3.9 (1.32)	3.79 (1.29)	<0.001
Gender, n (%)	Girl	1374 (51.33)	953 (48.60)	1201 (49.98)	513 (51.10)	0.29
	Boy	1303 (48.67)	1008 (51.40)	1202 (50.02)	491 (48.90)	
PA (h)		3.79 (2.11)	3.81 (2.21)	3.68 (2.16)	3.83 (2.26)	0.15
SB (h)		4.45 (2.71)	4.20 (2.60)	4.23 (2.68)	4.59 (2.60)	<0.001
SL (h)		11.15 (1.52)	11.18 (1.53)	10.96 (1.50)	11.17 (1.51)	<0.001
PedsQL (%)		76.80 (15.56)	76.79 (14.89)	78.40 (14.40)	77.57 (14.65)	<0.001

Note: Data are presented as the arithmetic mean ± SD unless otherwise indicated; PA = Physical activity, SL = Sleep, SB = Sedentary behaviour. PedsQL = Health-related quality of life, *p* = differences among four years estimated by ANOVA.

**Table 2 ijerph-19-14969-t002:** Compositional variation matrix for PA, SB, and SL from 2018–2021.

Year	Behaviour	PA	SB	SL
2018	PA	0.00	0.49	0.33
	SB	0.49	0.00	0.40
	SL	0.33	0.40	0.00
2019	PA	0.00	0.43	0.38
	SB	0.43	0.00	0.41
	SL	0.38	0.41	0.00
2020	PA	0.00	0.45	0.37
	SB	0.45	0.00	0.40
	SL	0.37	0.40	0.00
2021	PA	0.00	0.50	0.36
	SB	0.50	0.00	0.41
	SL	0.36	0.41	0.00
2018–2021	PA	0.00	0.49	0.36
	SB	0.49	0.00	0.41
	SL	0.36	0.41	0.00

PA = Physical activity, SB = Sedentary behaviour, SL = Sleep.

**Table 3 ijerph-19-14969-t003:** Adjusted linear regression models for QoL of preschoolers with PA, SB, and SL.

QoL of Years	Overall Composition		PA		SB		SL	
	R^2^	*p*	β_ilr_	*p*	β_ilr_	*p*	β_ilr_	*p*
2018	0.07	<0.001 ***	3.06	<0.001 ***	0.67	0.25	−2.39	<0.001 ***
2019	0.04	<0.001 ***	1.43	0.03 *	−1.16	0.09	−0.27	0.73
2020	0.03	<0.001 ***	−0.43	0.43	−1.56	0.008 **	2.00	0.004 **
2021	0.03	<0.001 ***	2.82	<0.001 ***	−2.61	0.002 **	−0.21	0.84

Model is adjusted for age, sex; β_ilr_ = isometric log-ratio regression coefficient of the specific behaviour relative to the other behaviours; PA = Physical activity; SL = Sleep; SB = Sedentary behaviour. * *p* < 0.05; ** *p* < 0.01; *** *p* < 0.001. Three decimals are retained because some of the *p*-values are very small.

**Table 4 ijerph-19-14969-t004:** Changes in QoL are predicted when reallocating time in PA, SB, and SL from the adjusted model in preschoolers.

Reallocated Time		Delta of QoL (BS:79.1)		
		From PA	From SB	From SL
15 min	to PA		0.24 (0.13,0.35) *	0.14 (0.04,0.23) *
	to SB	−0.24 (−0.35, −0.13) *		−0.10 (−0.18, −0.01) *
	to SL	−0.14 (−0.24, −0.05) *	0.10 (0.02, 0.19) *	
30 min	to PA		0.48 (0.26, 0.70) *	0.26 (0.08, 0.44) *
	to SB	−0.49 (−0.71, −0.26) *		0.19 (−0.36, −0.03) *
	to SL	−0.30 (−0.49, −0.10) *	0.21 (0.03, 0.39)	
45 min	to PA		0.71 (0.38, 1.05) *	0.39 (0.12, 0.65) *
	to SB	−0.74 (−1.08, −0.40) *		−0.28 (−0.53, −0.03) *
	to SL	−0.46 (−0.76, −0.16) *	0.33 (0.05,0.60) *	
60 min	to PA		0.95 (0.50,1.19) *	0.50 (0.15, 0.85) *
	to SB	−1.00 (−1.47, −0.53) *		−0.37 (−0.70, −0.04) *
	to SL	−0.63 (−1.05, −0.22) *	0.45 (0.08,0.82) *	

PA = Physical activity; SL = Sleep; SB = Sedentary behaviour; BS = Base line score; “from–to” expresses the transfer of time from one behaviour to another. 95% confidence interval of change in the QoL is calculated. * *p* < 0.05.

## Data Availability

Written permission for access to data from the corresponding author is required.

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
