# Peer review of "The Associations between 24-Hour Movement Behaviours and Quality of Life in Preschoolers: A Compositional Analysis of Cross-Sectional Data from 2018–2021"

_ijerph, 2022, doi:10.3390/ijerph192214969_

Round 1

Reviewer 1 Report

Congratulations on your work. I have the following comments:

* The innovative aspects of the work should be highlighted in the Introduction section. It is more detailed on page 13 but I missed this information at the beginning. At some point (page 2, lines 86-90) it seems that the only innovative aspect is the application of CODA and the focus on preschoolers.

* You should provide more details about CODA when describing the method.

Author Response

Please see the attachment for authors' responses to reviewer 1's comments.

Thank you for your comments.

Reviewer 2 Report

This is a very statistically sophisticated paper with interesting and important results.  There are some suggestions for improvement;

1)  The term "quality of life" should be defined in the first paragraph of the paper.

2)  The term "sedentary behaviors" should be defined in the second paragraph of the paper.

3)  In line 65, the nature of the "dose-effect relationship" should be clarified.

4)  In lines 71-72, the authors assert that quality of life improves with more sleep.  What happens with too much sleep?

5)  In line 72, what are "SL behavioral problems"?

6)  In line 75, the authors assert that PA, SL, and SB make up a 24-hour day of interdependent movement behaviors.  Are there other movement behaviors that occur during a 24-hour day?  If so, this should be acknowledged and potentially posed as a future research topic at the end of the paper.

7)  In the second paragraph of Section 2.1, the authors explain that data was collected at a much different time point in 2020 than in the other years.  This fact could have affected their results greatly.  They should discuss this as a major limitation in their Limitations section.

8)  In Section 2.2.1.5, the authors should give sample questions for PA, SB, and SL subscales.

9)  In Section 2.2.2, the authors state that there are two forms of the PedsQL, one for ages 2-4 and one for ages 5-6.  They then state that they use forms for ages 2-4 and 5-7.  Why do they add age 7?

10) The authors need to add sample items and report their own Cronbach's alpha for the PedsQL.

11) In Section 3.1, at the beginning, for the description of the sample, the authors should report the mean ages of parents and children, genders of parents, ethnicity of parents and children, and education and marital status of parents.

12) At the end of the first paragraph in Section 3.1, it is unclear what categories were compared for age and for differences in PA, SL, and SB.

13) In the first paragraph on the top of page 8, it would help the reader to see the variances reported in text and not just in the table.

14) For the paragraph directly under Table 3, more description of the different beta weights would be helpful to explain the U-shaped patterns instead of simply describing one beta weight for each pattern.

15) For the section "Trends in 24-hour movement behaviors of preschoolers over 2018-2021," explanation is given with literature support for the 2020 results, but literature support also needs to be given for the 2021 results.

16) In the Limitations paragraph, the authors mention the limitation of parents filling out all questionnaires.  Another issue with this is that this results in shared method variance.

17)  The authors need to mention the issue of generalizability of their study as a limitation beyond Singapore to other ethnicities, countries, and geographic areas.

Author Response

Thank you for your comments. Please see the attachment for authors' responses to reviewer 2's comments.

Reviewer 3 Report

How did You make sure parents did not complete the questionnaire twice?

What time-frame was the questionnaire referring to? Time of completion or before?

Shouldn’t You have attached the questionnaire as appendix C?

When stating "

the 4-year sample data showed significant differences in 285 age(Page0.05). For 24-hour behaviors (PA, SB and SL), 286 there were significant differences in SL and SB (PSL0.05) in the 4-year data sample" - which data series are You referring to?

"Further in 2021 and 2020, SB gradually  decreased and PA gradually increased in 2021" - rephrase

Figure 2 Group of QoL is missing for year 2019

Lines 360 and 451 have words with two fonts.

If collected date refers to moment of data collection, what is the importance of the circuit breaker if data was collected in august that year?

Shouldn't a seasonal variability of SL, PA and SB be taken into consideration?

Author Response

Dear reviewer 3, thank you for your comments. Please see the attachment for the authors' responses to reviewer 3.

Round 2

Reviewer 2 Report

All concerns have been addressed adequately.  The paper is ready to be published.